# Monitoring of Nitrogen Concentration in Soybean Leaves at Multiple Spatial Vertical Scales Based on Spectral Parameters

**DOI:** 10.3390/plants13010140

**Published:** 2024-01-04

**Authors:** Tao Sun, Zhijun Li, Zhangkai Wang, Yuchen Liu, Zhiheng Zhu, Yizheng Zhao, Weihao Xie, Shihao Cui, Guofu Chen, Wanli Yang, Zhitao Zhang, Fucang Zhang

**Affiliations:** 1Key Laboratory of Agricultural Soil and Water Engineering in Arid and Semiarid Areas of Ministry of Education, Northwest A&F University, Xianyang 712100, China; 2021050986@nwsuaf.edu.cn (T.S.); wangzhk2023@163.com (Z.W.); 18729390136@163.com (Y.L.); guci_owo@163.com (Z.Z.); zhao24989@nwafu.edu.cn (Y.Z.); 2021012220@nwsuaf.edu.cn (W.X.); 2022012353@nwafu.edu.cn (S.C.); 2022012367@nwafu.edu.cn (G.C.); 2022012387@nwafu.edu.cn (W.Y.); zhangzhitao@nwafu.edu.cn (Z.Z.); zhangfc@nwsuaf.edu.cn (F.Z.); 2Institute of Water-Saving Agriculture in Arid Areas of China, Northwest A&F University, Xianyang 712100, China

**Keywords:** soybean, remote sensing, hyperspectral, leaf nitrogen content, spectral parameters

## Abstract

Nitrogen is a fundamental component for building amino acids and proteins, playing a crucial role in the growth and development of plants. Leaf nitrogen concentration (LNC) serves as a key indicator for assessing plant growth and development. Monitoring LNC provides insights into the absorption and utilization of nitrogen from the soil, offering valuable information for rational nutrient management. This, in turn, contributes to optimizing nutrient supply, enhancing crop yields, and minimizing adverse environmental impacts. Efficient and non-destructive estimation of crop LNC is of paramount importance for on-field crop management. Spectral technology, with its advantages of repeatability and high-throughput observations, provides a feasible method for obtaining LNC data. This study explores the responsiveness of spectral parameters to soybean LNC at different vertical scales, aiming to refine nitrogen management in soybeans. This research collected hyperspectral reflectance data and LNC data from different leaf layers of soybeans. Three types of spectral parameters, nitrogen-sensitive empirical spectral indices, randomly combined dual-band spectral indices, and “three-edge” parameters, were calculated. Four optimal spectral index selection strategies were constructed based on the correlation coefficients between the spectral parameters and LNC for each leaf layer. These strategies included empirical spectral index combinations (Combination 1), randomly combined dual-band spectral index combinations (Combination 2), “three-edge” parameter combinations (Combination 3), and a mixed combination (Combination 4). Subsequently, these four combinations were used as input variables to build LNC estimation models for soybeans at different vertical scales using partial least squares regression (PLSR), random forest (RF), and a backpropagation neural network (BPNN). The results demonstrated that the correlation coefficients between the LNC and spectral parameters reached the highest values in the upper soybean leaves, with most parameters showing significant correlations with the LNC (*p* < 0.05). Notably, the reciprocal difference index (VI6) exhibited the highest correlation with the upper-layer LNC at 0.732, with a wavelength combination of 841 nm and 842 nm. In constructing the LNC estimation models for soybeans at different leaf layers, the accuracy of the models gradually improved with the increasing height of the soybean plants. The upper layer exhibited the best estimation performance, with a validation set coefficient of determination (R^2^) that was higher by 9.9% to 16.0% compared to other layers. RF demonstrated the highest accuracy in estimating the upper-layer LNC, with a validation set R2 higher by 6.2% to 8.8% compared to other models. The RMSE was lower by 2.1% to 7.0%, and the MRE was lower by 4.7% to 5.6% compared to other models. Among different input combinations, Combination 4 achieved the highest accuracy, with a validation set R^2^ higher by 2.3% to 13.7%. In conclusion, by employing Combination 4 as the input, the RF model achieved the optimal estimation results for the upper-layer LNC, with a validation set R^2^ of 0.856, RMSE of 0.551, and MRE of 10.405%. The findings of this study provide technical support for remote sensing monitoring of soybean LNCs at different spatial scales.

## 1. Introduction

Soybean, as one of the world’s top five crops, serves as a crucial source of high-quality protein and edible oil for humanity and plays a vital role in global food security [1]. In China, the soybean planting area accounts for approximately 7.7% of the world’s total, yet its yield is only 70.7% of the world average [2]. Ensuring high soybean yields is imperative. Nitrogen is a fundamental component of plant proteins, nucleic acids, amino acids, and other essential biomolecules, playing a key role in plant growth and development [3,4]. Monitoring leaf nitrogen concentration (LNC) allows an assessment of whether plants receive sufficient nitrogen supply [5,6], facilitating the adoption of appropriate measures to promote or adjust plant growth. Simultaneously, adequate nitrogen directly influences crop yield and quality. Proper fertilization and monitoring of LNCs assists farmers in effectively managing nitrogen supply, avoiding over- or under-fertilization, thereby enhancing crop productivity and quality [7,8]. Moreover, excessive nitrogen fertilization can lead to nitrogen leakage into soil and water, causing environmental issues such as nutrient enrichment of water bodies and soil acidification [9,10]. Monitoring LNCs enables precise determination of plant nitrogen requirements, helping to reduce over-fertilization and mitigate environmental impacts. Hence, the importance of monitoring crop LNCs is evident [11,12].

Traditional laboratory methods for collecting leaf samples and detecting LNC, such as the Kjeldahl method [13,14], are characterized by high sensitivity and accuracy. However, these methods involve destructive sampling, resulting in complex sample handling, low efficiency, and high costs. This approach may potentially damage plants, adversely affecting crop growth and development [15,16]. Efficient and non-destructive monitoring of crop growth is central to modern precision agriculture [17]. Current manual methods for measuring LNC suffer from limitations, including small measurement areas, high workload, and poor data representativeness, making them unsuitable for large-scale precision management of field crops [18]. With the widespread application of remote sensing technology in agriculture, timely and non-destructive monitoring of crop LNCs has become feasible [19,20].

Spectral remote sensing technology, with its ability to provide rich spectral information and conduct large-scale non-contact monitoring, has gained attention in the study of crop physiological growth indicators [21,22]. Numerous studies have been conducted on using spectral remote sensing technology to monitor crop LNC [23,24,25]. For example, Fan et al. (2019) [23] employed a continuous projection algorithm to select spectral parameters with optimal performance for LNC estimation, achieving a verification set determination coefficient (R^2^) exceeding 0.75, indicating excellent estimation accuracy. Zhao et al. (2021) [24], through field experiments over four growing seasons, developed a novel method based on remote sensing data to calculate nitrogen parameters for winter wheat. The model exhibited good stability. Shu et al. (2023) [25], using high-resolution spectral imaging obtained from unmanned aerial vehicles, estimated the nitrogen status of corn leaves using spectral decomposition methods, significantly improving the accuracy of nitrogen status estimation in corn leaves based on unmanned aerial vehicle high-resolution spectral images. These studies suggest that spectral information has the capability to monitor crop LNC, and although constructing fixed-wavelength spectral indices can accurately monitor nitrogen conditions to some extent, the different physiological information of crops due to factors such as their growth environment and growth stage can lead to variations in their spectral characteristics [26,27]. In such cases, using the same wavelengths may result in inadequate utilization of spectral data, limiting the effectiveness of the calculated spectral index inversion model and reducing model accuracy [28]. Furthermore, for crops, there is a lack of comparison and discussion on the prediction effects of different vertical scales of LNCs. Models are often built and analyzed using LNCs from a specific site without comparison and optimization, which limits the spatial applicability of the established prediction models [29].

To address these issues, this study aims to construct three categories of spectral parameters for estimating LNCs at various soybean leaf layers: (1) empirical spectral indices with good correlations to crop parameters from previous studies; (2) optimal spectral indices, i.e., the best combination of indices within the wavelength range of 350–1830 nm with the highest correlation to the LNC at various soybean leaf layers; and (3) three-edge spectral indices, involving blue, yellow, and red edge areas. These parameters, often associated with red, blue, and green edge areas, provide valuable insights into the spectral characteristics of the studied vegetation. The four-node stage (V4) of soybeans, occurring when the fourth true leaf unfolds after planting, is a critical period for soybean growth and development [7]. During this stage, plants begin rapid growth and establish initial structures, such as leaves and stems. The health of plants at this stage directly affects subsequent growth and development. Additionally, soybeans require sufficient nutrients to support their growth and development at this stage. Therefore, monitoring LNC during this period is of paramount importance for field management to ensure soybeans receive the necessary nutrition for stable and high yields. In this study, different soybean leaf layers’ LNCs under various treatments at the V4 stage were selected as the research objects. We constructed different types of spectral parameters and established soybean LNC estimation models based on the partial least squares regression (PLSR), random forest (RF), and backpropagation neural network (BPNN) algorithms. We compared and analyzed the estimation effects and stability of the models, aiming to establish LNC prediction models at different vertical scales for soybeans to achieve non-destructive and rapid LNC estimation.

## 2. Materials and Methods

### 2.1. Research Area and Test Design

The experiment was carried out at the water-saving irrigation experimental station (34°14′ N, 108°10′ E, altitude 521 m) of the Water-Saving Agriculture Research Institute of Northwest A&F University (Figure 1) in June–September 2021 and June–September 2022. In the experiment, 24 experimental plots were set up, each of which was 6 m long and 4 m wide. The experiment set four nitrogen application levels, 0 kg ha^−1^ (N0), 60 kg ha^−1^ (N1), 120 kg ha^−1^ (N2), and 180 kg ha^−1^ (N3), and two seed dressing treatments, rhizobium inoculation (R) and water seed dressing (unmarked). The experiment was conducted in a completely randomized design with three replicates. In order to reduce the influence between experimental treatments, a 2 m wide isolation belt was set between adjacent cells. The amount of phosphate and potassium fertilizer in each experimental plot was 30 kg ha^−1^. The nitrogen fertilizer used in the experiment was urea (46% N), the phosphorus fertilizer was calcium superphosphate (16% P), and the potassium fertilizer was potassium chloride (62% K).

The seed dressing method was used to inoculate rhizobium (in line with the national industry standard GB20287-2006 [30]). The seed dressing method was used and 25 g of rhizobium powder was added into 500 ml water and stirred evenly. Before soaking, the rhizobium was fully shaken to ensure that the rhizobium was evenly attached to the seed surface. The seeds were dried after seed dressing, and the dried seeds were sown on 18 June 2021 and 10 June 2022, respectively. The planting density was 300,000 plants ha^−1^, the row spacing was 50 cm, and the plant spacing was 10 cm. Soybeans were harvested on 30 September 2021 and 20 September 2022, respectively.

### 2.2. Data Collection

The LNC and hyperspectral data were obtained in the experimental plots at V4 (17 July 2021 and 15 July 2022). In the two-year experiment, 48 groups of leaf nitrogen concentration and hyperspectral reflectance samples were obtained, respectively. After removing the outliers, there were 48 sets of data samples, and 2/3 of the samples were selected as the modeling set, with the remainder of the samples used as the validation set.

#### 2.2.1. Measurement of Leaf Nitrogen Concentration

In our experiment, the root leaf nitrogen concentration (N_RL_), lateral leaf nitrogen concentration (N_LL_) and canopy leaf nitrogen concentration (N_CL_) of the same soybean plant were collected at the same time, and the measurement sites are shown in Figure 2. In each plot, 10 soybean plants were randomly selected to determine their nitrogen concentration, and the average value was used to represent the nitrogen concentration value of different leaf layers in the plot. These leaves were dried in an oven at 105 °C for 30 min and then extra dried at 75 °C until a constant weight was achieved. Then, the dried samples were ground through a 1 mm sieve, digested with H_2_SO_4_–H_2_O_2_, and LNC was analyzed by the Kjeldahl method. The detailed determination process used Cheng et al. (2022) as a reference [13].

#### 2.2.2. Acquisition of Hyperspectral Data

The spectral acquisition instrument used was a FieldSpec3 hyperspectrometer produced by the ASD company in the United States. The wavelength range of the instrument was 350~1830 nm. The spectral resolution of 350~1000 nm was 3 nm, and the sampling interval was 1.4 nm. The resolution of 1000~1830 nm was 10 nm, and the sampling interval was 2 nm. The instrument automatically interpolated the sampling data into a 1 nm interval output. The fiber length was 1.5 m and the field of view was 25°. The determination was carried out at 11: 00–13: 00 in sunny and windless weather. During the measurement, the spectrometer probe was about 15 cm away from the soybean canopy, always keeping 90° with the ground, and the field angle was 25°. Before the spectral determination, the instrument was corrected by a diffuse reflection reference plate with a reflectivity of 1. The instrument was optimized every 5 min, and the dark current was collected every 5 min to optimize the instrument. In each plot, the soybean around the sample point was measured by five-point plum blossom sampling, and the average value was taken as the final spectral value of the monitoring point. To reduce or eliminate the influence of useless information such as background noise, baseline drift, and stray light on the spectral reflectance curve, we used Savitzky–Golay convolution smoothing (9 points and 4 times) to preprocess the spectral data [2].

### 2.3. Techniques for Data Analysis

The spectral index can reflect crop growth and nutritional status [26,27]. In this study, three types of spectral parameters were constructed to estimate the soybean’s LNC: (1) the empirical spectral index with good correlation between previous studies and crop parameters; (2) the optimal spectral index with the highest correlation with the soybean’s LNC was selected, that is, the best combination index in the range of 350–1830 nm; and (3) “trilateral” spectral parameters such as blue, yellow, and red edge areas. The selected spectral parameters were calculated as shown in Table 1. The calculation results of the spectral index were calculated by MATLAB R2022 (MathWorks, Inc., Natick, MA, USA). All the figures in this study were drawn by Origin Pro 2021 (OriginLab Corp., Northampton, MA, USA).

Firstly, the correlation between the spectral parameters of the soybean leaves and LNCs in all 24 plots was analyzed, and four screening strategies were designed: (1) Combination 1, the empirical spectral indices with significant correlation coefficients (*p* < 0.05) with the LNC of each leaf layer were selected; (2) Combination 2, the spectral indices with significant correlation coefficients (*p* < 0.05) with the LNC of each leaf layer were selected from the random combination of dual-band spectral indices; (3) Combination 3, the spectral parameters that were significant (*p* < 0.05) to the LNC of each leaf layer were selected from the “trilateral” spectral parameters; and (4) Combination 4, all spectral parameters of Combination 1, Combination 2, and Combination 3 with significant correlation coefficients (*p* < 0.05) with the LNC of each leaf layer were selected as input variables of the model.

Subsequently, we used MATLAB R2022 to test three machine learning methods, namely PLSR, RF, and BPNN. In the construction of the PLSR model, a 10-fold cross-validation method was employed to determine the optimal number of latent variables (LVs) for estimating the LNC. In this study, the optimal number of latent variables for the LNC estimation model was determined to be 3. In the construction of the RF model, after parameter optimization and multiple training, the number of decision trees in the LNC model was set to 600. The hidden layer transfer function of the BPNN was set to “TANSIG”, and the Levenberg–Marquardt (trainlm) algorithm based on numerical optimization theory was used as the network training function. After much training, the number of neurons in the middle layer was determined to be 15.

### 2.4. Techniques for Data Analysis

In order to verify the prediction accuracy and predictive ability of the models, three indicators were selected, the determination coefficient (R^2^), root mean square error (RMSE), and mean relative error (MRE), to evaluate the model accuracy [6]. The R^2^, RMSE, and MRE were calculated using the following equations:(1)$$R^{2}=\frac{\sum_{\dot{i}=1}^{n}{\left({\hat{y}}_{i}-\bar{y}\right)}^{2}}{\sum_{i=1}^{n}{\left(y_{i}-\bar{y}\right)}^{2}}$$
(2)$$\mathrm{RMSE}=\sqrt{\frac{\sum_{i=1}^{n}{\left({\hat{y}}_{i}-y_{i}\right)}^{2}}{n}}$$
(3)$$\mathrm{MRE}=\frac{1}{n}\overset{n}{\underset{i=1}{\sum }}\frac{\left|{\hat{y}}_{i}-y_{i}\right|}{y_{i}}\times 100\%$$

## 3. Results

### 3.1. Comparison of LNCs in Different Leaf Layers of Soybean and Division of Sample Set

The crops’ LNCs exhibited spatial and vertical heterogeneity [27]. The statistical results of the LNC are depicted in Figure 3 and Table 2. It can be observed that the soybean leaf nitrogen concentration followed the size order of N_RL_ < N_LL_ < N_CL_, indicating a gradual increase in LNC from the root system to the top of the soybean plant.

From Table 2, it is evident that different nitrogen application rates significantly impacted the LNCs of the soybeans (*p* < 0.05). In the majority of leaves, the inoculation of rhizobia also exerted a significant effect on the LNC (*p* < 0.05). Notably, as nitrogen application increased, leaf LNC increased accordingly. Furthermore, under the condition of the same nitrogen application rate, rhizobia inoculation promoted nitrogen absorption in the leaves, thereby augmenting the LNC.

### 3.2. Correlation Analysis between Spectral Parameters and LNCs of Soybean Leaf Layers

The correlation analysis results between the empirical spectral indices, “three-side parameters”, and LNCs at different leaf layers are presented in Table 3. The process of wavelength combination selection for the calculated indices from pairwise spectral bands is detailed in Figure 4. The correlation coefficients and wavelength positions of the ten selected indices with different leaf layers are shown in Figure 4 and Table 4. The results indicate that the majority of “three-side parameters”, arbitrary two-band spectral indices, and empirical spectral indices exhibited higher correlation coefficients with the N_CL_ compared to the N_LL_ and N_RL_. Among the indices constructed with two bands in the N_CL_, the highest correlation with the LNC was achieved, with all exceeding 0.65. Notably, the index VI6 demonstrated the highest correlation with N_CL_ at 0.732, with a wavelength combination of 841 nm and 842 nm. Simultaneously, most of the selected spectral parameters in this study showed a significant correlation level (*p* < 0.05) with the LNC at various leaf layers. For the empirical indices, the index D833/658 had the highest correlation with N_CL_ at 0.671, while the “three-side parameter” Sr had the highest correlation with N_CL_ at 0.669. In general, the correlation coefficient between the spectral parameters and LNC gradually increased from the root to the top of the soybean leaves.

### 3.3. Construction of Soybean LNC Estimation Model at a Multi-Spatial Vertical Scale

In the preceding section, we computed the correlation coefficients between the spectral parameters and soybean LNCs at various layers. We selected different types of spectral parameters as inputs for the machine learning model, namely, Combination 1 (IPVI, OSAVI, NDNI, AVI, D_678/500_, D_800/550_, D_800/680_, D_833/658_, DVI_MSS_, and DDI), Combination 2 (D_b_, D_y_, D_r_, R_g_, S_b_, and S_r_), Combination 3 (RI, DI, SAVI, NDVI, TVI, mSR, mNDI, PI, SI, and VI6), and Combination 4 (IPVI, OSAVI, NDNI, AVI, D_678/500_, D_800/550_, D_800/680_, D_833/658_, DVI_MSS_, D_D_, D_b_, D_y_, D_r_, R_g_, S_b_, S_r_, RI, DI, SAVI, NDVI, TVI, mSR, mNDI, PI, SI, and VI6). The selected spectral parameters for each combination were consistent across different leaf layers. Subsequently, using the aforementioned combinations as independent variables and the soybean LNCs at various layers as the response variables, we employed PLSR, RF, and BPNN to construct soybean LNC estimation models. Model accuracy was comprehensively evaluated based on R^2^. The prediction results for the soybean leaf areas by different modeling methods are illustrated in Figure 5, Figure 6 and Figure 7. The results indicate that, with the increase in the vertical height of soybean, the accuracy of the estimation model gradually improved. The N_cl_ exhibited the best estimation performance, with a validation set R^2^ higher by 9.9% to 16.0% compared to other layers under similar conditions. Analyzing the estimation model for N_cl_, random forest (RF) demonstrated the highest accuracy in estimating LNC. The validation set R^2^ was 6.2% to 8.8% higher than other models, RMSE was 2.1% to 7.0% lower, and MRE was 4.7% to 5.6% lower than other models. Considering different input combinations, Combination 4 as the input achieved the highest accuracy, with a validation set R^2^ higher by 2.3% to 13.7% compared to other models. In summary, using Combination 4 as the input, the RF model produced the optimal estimation results for N_CL_, with a validation set R^2^ of 0.856, RMSE of 0.551, and MRE of 10.405%.

## 4. Discussion

Spectral information serves as the foundational data for the optical remote sensing of spectral traits [46,47], and it is determined by various factors such as canopy structure and biochemical composition. Due to the uneven distribution of nutrients along the vertical scale of crops, it is often manifested on the leaves [29]. Studying the vertical variation in LNC can better elucidate changes in plant physiological processes and adjust the photosynthesis and nutrient absorption of crops, thereby optimizing agricultural management practices more effectively.

This study revealed that with the deepening of vertical spatial scale (closer to the root), LNC gradually decreased. This may be attributed to the fact that the growth point of crops is usually located at the top of the plant or at the top of its branches, which is the primary region for new tissue development. New tissues have a higher demand for nitrogen as it is a crucial component in building proteins and other biomolecules [48]. Simultaneously, upper leaves typically receive more sunlight, making them more exposed to sunlight [49]. Photosynthesis, the process through which plants synthesize organic compounds using sunlight, has a high demand for nitrogen [50]. Therefore, to meet the needs of photosynthesis, upper leaves may have a relatively higher nitrogen concentration to support the development of new tissues. This study also found that when predicting crop leaf LNCs on the vertical scale using spectral parameters, the prediction performance was relatively better for upper leaves, while the prediction for middle and lower leaves was poorer. This is because the ability of light to penetrate through crop plants may significantly differ at different vertical levels. Upper leaves are usually directly exposed to sunlight, making them easier to identify for spectral sensors. In contrast, middle and lower leaves may be shaded by upper leaves, resulting in poor light transmission [51]. This may reduce the quality of observations for these leaves, and the spectral signals at different vertical levels may mix together, especially when the sensor resolution is relatively low. This can make it challenging to differentiate the spectral signals at different vertical levels, and in general, upper leaves usually have more sensitive spectral signals, making them easier to detect and interpret, while the signal mixing at the middle and lower layers may complicate the estimation of LNC [29]. We observed that the increase in nitrogen application led to a corresponding rise in LNC. This is attributed to the heightened nitrogen application, which enhances soybean’s capacity to absorb nitrogen, consequently elevating LNC [7]. Simultaneously, our findings indicate that inoculation with rhizobia enhanced soybean LNC. This is attributed to the symbiotic relationship between rhizobia and soybeans, wherein rhizobia facilitate nitrogen fixation, thereby promoting nitrogen absorption [52].

Through the correlation analysis of different spectral parameters with LNC at different leaf layers, it was found that randomly selected two-band spectral indices had higher correlation coefficients with the LNC at different leaf layers compared to empirical spectral indices and three-side parameters. This is because, for different study subjects, due to variations in growth environments, growth stages, and other factors, the physiological information of crops may differ, leading to different spectral characteristics. In such cases, using empirically calculated spectral parameters with the same wavelengths may underutilize spectral data [26,27,28]. Additionally, three-side spectral parameters may not sufficiently extract information related to nitrogen concentration in plant leaves. Conversely, randomly selected spectral indices may more effectively capture this information by searching for correlations between randomly chosen wavelengths [53].

The accuracy of the soybean LNC estimation models varied significantly based on the different model inputs and machine learning methods. When constructing soybean LNC estimation models using different input combinations, the soybean LNC estimation model based on random forest (RF) had the highest accuracy. This suggests that RF is more advantageous in estimating soybean LNC compared to other models, which is consistent with previous results for crop physiological growth indicators [28,53]. This is because RF is an ensemble learning method based on decision trees, which can effectively capture nonlinear relationships. The relationship between the LNC and spectral indices may be a complex nonlinear relationship, and RF may be more suitable for handling such nonlinear relationships [54]. Additionally, random forests typically exhibit good resistance to overfitting, which is helpful when dealing with small sample data or noise in the data. RF introduces randomness by building multiple decision trees and combining their results, reducing the risk of overfitting [53]. In contrast, when there are a large number of highly correlated predictor variables, partial least squares regression (PLSR) adapts to fewer components by considering the dependent variable, reducing the estimation accuracy of the model [55]. However, the backpropagation neural network (BPNN) has limitations due to the selection of model parameters, and too many feature parameters can lead to computational redundancy and insufficient timeliness, resulting in reduced accuracy [56]. Notably, when the machine learning models were the same, we found that the accuracy of the model input with Combination 2 (randomly selected two-band spectral indices) was lower than that with Combination 4 (various spectral parameters selected). This finding can be attributed to Combination 4 incorporating features from Combinations 1–3, maximizing the extraction of hyperspectral information. The lower accuracy of the models built with Combinations 1 to 3 as inputs was due to their inherent limitations as empirical parameters, as explained earlier. Similarly, it is not difficult to explain that the estimation of N_cl_ is better under the same conditions, as hyperspectral collection is more sensitive to the upper canopy.

This study aimed to estimate soybean LNC in vertical layers using spectral parameters calculated from drone multispectral image information. As soybean field management has significant practical implications, further research is needed. Future studies could consider developing more precise and accurate new indices rather than relying solely on conventional indices. Additionally, the use of drone hyperspectral imaging and thermal infrared imaging will be considered. Furthermore, spectral sensors will be adjusted multiple times in terms of angles and heights to capture more spectral information of soybean plants in three-dimensional spaces, achieving precise monitoring of physiological data for lower leaves and ultimately improving the estimation accuracy of soybean LNC and enhancing soybean yield.

## 5. Conclusions

This study, based on plot experiments and hyperspectral measured data, estimated the nitrogen concentration values in different leaf layers of soybeans. This was achieved by constructing spectral parameters, including empirical spectral indices, spectral indices selected from any two bands, and “three-edge” parameters. Three machine learning models, namely PLSR, RF, and BPNN, were employed for estimation. The conclusions drawn were the following: the majority of spectral parameters showed significant correlations with soybean leaf nitrogen concentration (LNC) at a significance level of *p* < 0.05. As soybean height increased, the correlation coefficients between LNC and spectral parameters also increased. Among them, VI6 exhibited the highest correlation with upper-layer leaf nitrogen concentration, reaching 0.732, with a wavelength combination of 841 nm and 842 nm. In summary, using Combination 4 (spectral parameters significantly correlated (*p* < 0.05) with leaf LNC) as the input, the RF model yielded the optimal estimation results for the upper-layer LNC, with a validation set R^2^ of 0.856, RMSE of 0.551, and MRE of 10.405%.

## Figures and Tables

**Figure 1 plants-13-00140-f001:**
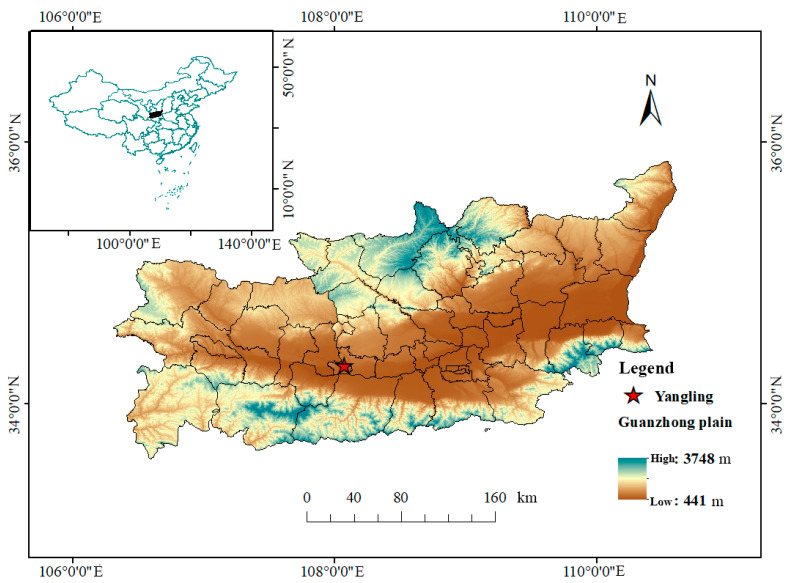
Study area.

**Figure 2 plants-13-00140-f002:**
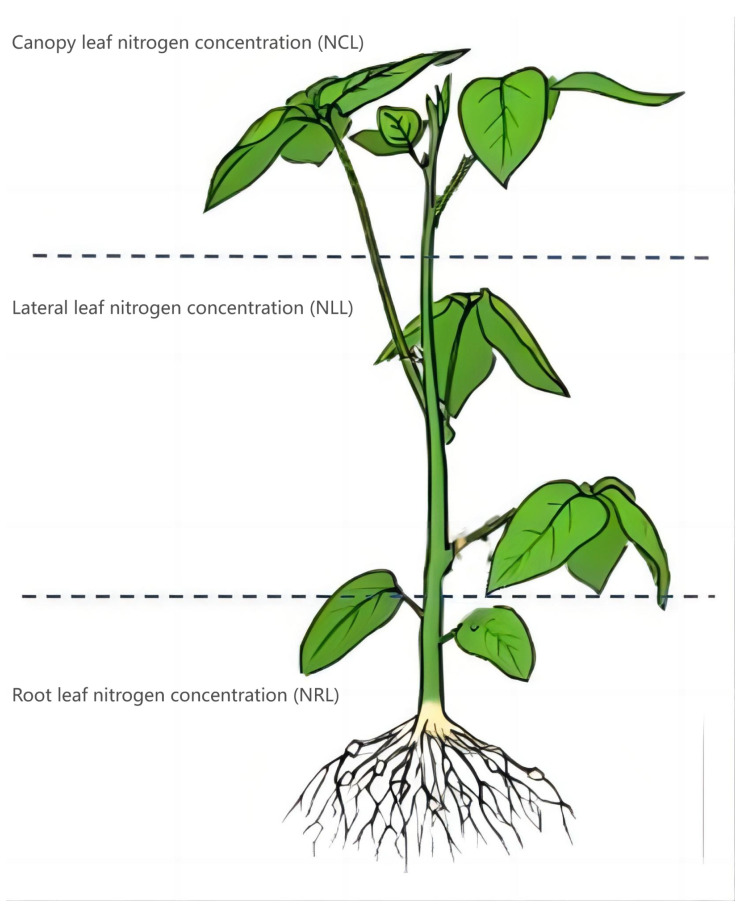
Leaf sampling details of each layer of the soybean plant.

**Figure 3 plants-13-00140-f003:**
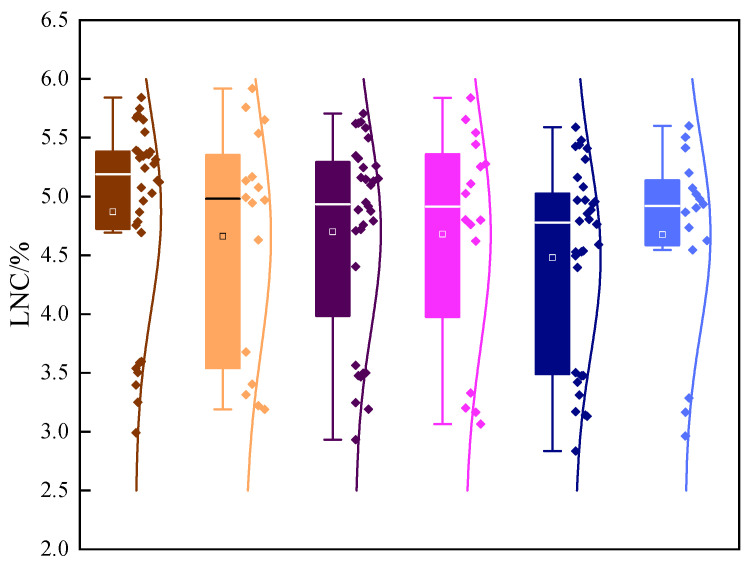
Statistics of the LNC in each leaf layer of soybean. The horizontal line in the box line diagram represents the median, and the white box represents the average value. The N_RL_ modeling set is dark brown and the validation set is light brown. The N_LL_ modeling set is dark purple and the validation set is light purple. The N_CL_ modeling set is dark blue and the validation set is light blue.

**Figure 4 plants-13-00140-f004:**
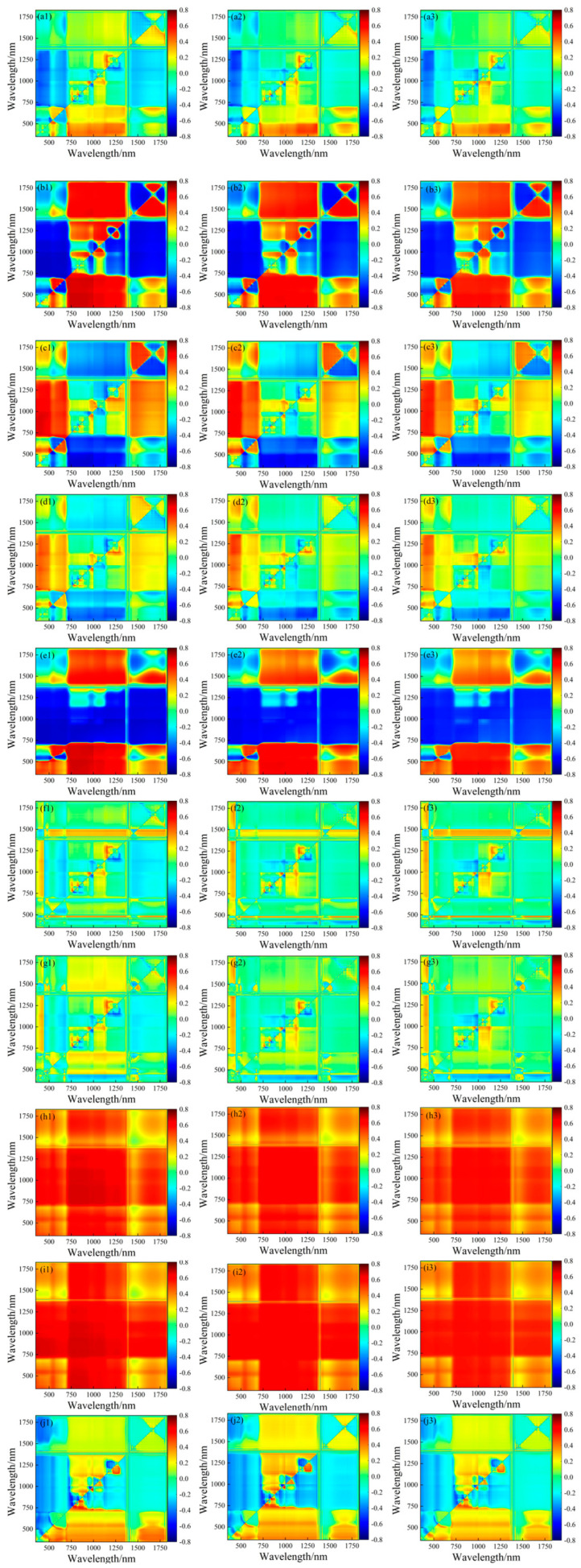
The correlation matrix diagrams of the spectral indices and soybean LNCs. (**a1**) RI and N_CL_; (**a2**) RI and N_LL_; (**a3**) RI and N_RL_; (**b1**) DI and N_CL_; (**b2**) DI and N_LL_; (**b3**) DI and N_RL_; (**c1**) SAVI and N_CL_; (**c2**) SAVI and N_LL_; (**c3**) SAVI and N_RL_; (**d1**) NDVI and N_CL_; (**d2**) NDVI and N_LL_; (**d3**) NDVI and N_RL_; (**e1**) TVI and N_CL_; (**e2**) TVI and N_LL_; (**e3**) TVI and N_RL_; (**f1**) mSR and N_CL_; (**f2**) mSR and N_LL_; (**f3**) mSR and N_RL_; (**g1**) mNDI and N_CL_; (**g2**) mNDI and N_LL_; (**g3**) mNDI and N_RL_; (**h1**) PI and LNC_CL_; (**h2**) PI and LNC_LL_; (**h3**) PI and LNC_RL_; (**i1**) SI and LNC_CL_; (**i2**) SI and LNC_LL_; (**i3**) SI and LNC_RL_; (**j1**) VI6 and LNC_CL_; (**j2**) VI6 and LNC_LL_; and (**j3**) VI6 and LNC_RL_. The colors from blue to red represent the negative correlation to positive correlation.

**Figure 5 plants-13-00140-f005:**
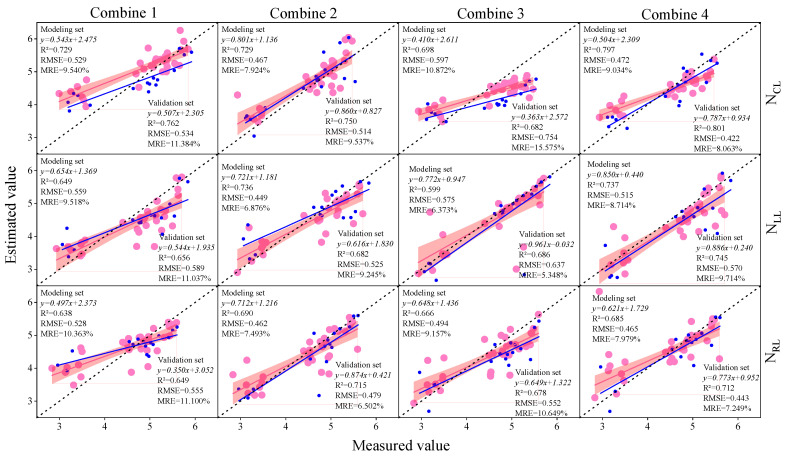
The modeling set and validation sets of the BPNN estimation models with different input variables and leaf layers. The red dots and red lines represent the modeling sets and the modeling set fitted curves, the blue dots and blue lines represent the verification sets and the verification sets fitted curve, and the dotted lines represent the 1:1 lines.

**Figure 6 plants-13-00140-f006:**
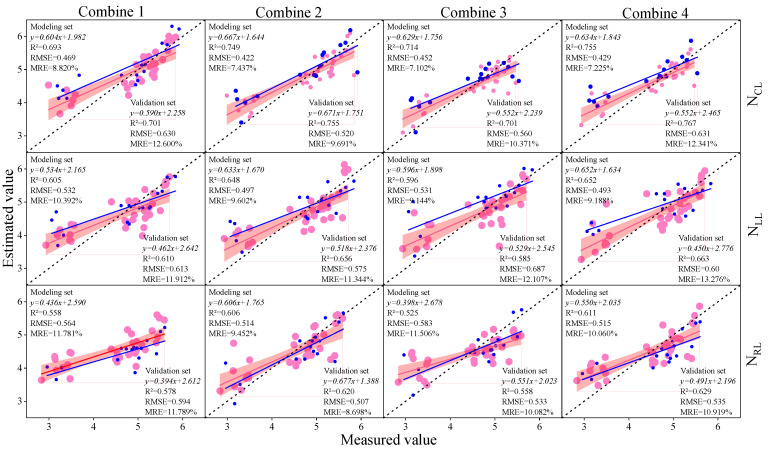
The modeling sets and validation sets of the PLSR estimation models with different input variables and leaf layers. The red dots and red lines represent the modeling sets and the modeling set fitted curves, the blue dots and blue lines represent the verification sets and the verification set fitted curves, and the dotted lines represent the 1:1 lines.

**Figure 7 plants-13-00140-f007:**
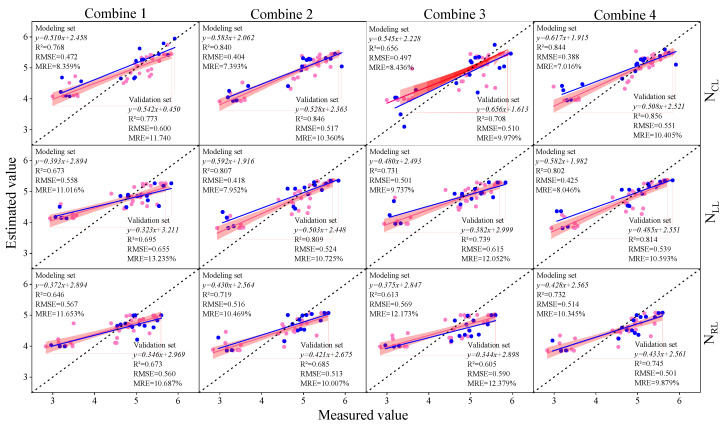
The modeling sets and validation sets of RF estimation models with different input variables and leaf layers. The red dots and red lines represent the modeling sets and the modeling set fitted curves, the blue dots and blue lines represent the verification sets and the verification set fitted curves, and the dotted lines represent the 1:1 lines.

**Table 1 plants-13-00140-t001:** Selection of spectral parameters, calculation formulas, and reference source.

Selected Spectral Parameter	Calculation Formula	Reference
Maximum first-order derivative value in the blue edge (490–530 nm) *D_b_*	–	[31]
Maximum first-order derivative value in the yellow edge (462–642 nm) *D_y_*	–	[31]
Maximum first-order derivative value in the red edge (670–760 nm) *D_r_*	–	[32]
Maximum reflectivity of the green peak (510–560 nm) *R_g_*	–	[33]
Minimum reflectivity of the red valley (650–690 nm) *R_r_*	–	[33]
Blue edge (490–530 nm) area *S_b_*	Sum of first-order derivatives within the blue edge wavelength range	[31]
Yellow edge (462–642 nm) area *S_y_*	Sum of first-order derivatives within the yellow edge wavelength range	[31]
Red edge (670–760 nm) area *S_r_*	Sum of first-order derivatives within the red edge wavelength range	[32]
Normalized red-blue amplitude difference (NDDr.b)	(*D_r_* − *D_b_*)/(*D_r_* + *D_b_*)	[34]
Normalized first-order red-blue amplitude difference (NDSDr.b)	(*SD_r_* − *SD_b_*)/(*SD_r_* + *SD_b_*)	[34]
Infrared percentage vegetation index (IPVI)	*R*_800_ × (*R*_800_ + *R*_670_)	[35]
Optimized soil-adjusted vegetation index (OSAVI)	$(1+0.16)(R_{800}-R_{670})/\left(R_{800}+R_{670}+0.16\right)$	[36]
Normalized difference nitrogen index (NDNI)	$\frac{㏒\left(\frac{1}{R_{1510}}\right)-㏒\left(\frac{1}{R_{1680}}\right)}{㏒\left(\frac{1}{R_{1510}}\right)+㏒\left(\frac{1}{R_{1680}}\right)}$	[37]
Ashburn vegetation index (AVI)	$2\times \frac{R_{800}}{R_{1100}}-\frac{R_{600}}{R_{700}}$	[38]
Difference 678/500 (D678/500)	$R_{678}/R_{500}$	[39]
Difference 800/550 (D800/550)	$R_{800}/R_{550}$	[40]
Difference 800/680 (D800/680)	$R_{800}/R_{680}$	[41]
Difference 833/658 (D833/658)	$R_{833}/R_{658}$	[42]
Differenced vegetation index MSS (DVIMSS)	$2.4\times \frac{R_{800}}{R_{1100}}-\frac{R_{600}}{R_{700}}$	[43]
Double difference index (DD)	$(R_{749}-R_{720)}-\left(R_{701}-R_{672}\right)$	[44]
Ratio index (RI)	$R_{i}/R_{j}$	[27]
Difference index (DI)	$R_{i}$ $-R_{j}$	[27]
Soil-adjusted vegetation index (SAVI)	$\left(1+0.16\right)\frac{R_{i}-R_{j}}{R_{i}+R_{j}+0.16}$	[28]
Normalized difference vegetation index (NDVI)	$R_{i}-R_{j}/R_{i}+R_{j}$	[28]
Triangular vegetation index (TVI)	$0.5\times \left(120\times \left(R_{i}-R_{550}\right)-200\times \left(R_{j}-R_{550}\right)\right)$	[28]
Modified simple ratio (mSR)	$R_{i}-R_{455}/R_{j}-R_{455}$	[28]
Modified normalized difference index (mNDI)	$R_{i}-R_{j}/R_{i}+R_{j}$ $-2R_{455}$	[28]
Product index (PI)	$R_{i}\times R_{j}$	[45]
Sum index (SI)	$R_{i}+R_{j}$	[45]
Reciprocal difference index (VI6)	$1/R_{i}$ $-1/R_{j}$	[45]

Note: $R_{i}$ (*i* = 1,2,3) is the reflectivity at any band, and R445, R455, R500, R530, R531, R550, R570, R670, R680, R700, R705, R742, R750, and R800 represent the spectral reflectance of wavelengths 445 nm, 455 nm, 500 nm, 530 nm, 531 nm, 550 nm, 570 nm, 670 nm, 680 nm, 700 nm, 705 nm, 742 nm, 750 nm, and 800 nm.

**Table 2 plants-13-00140-t002:** The values of the LNCs in each leaf layer under different treatments. R, the influence of different rhizobium inoculation methods on each index; N, the effect of different nitrogen applications on each index; N*R, the influence of the interaction between the rhizobium inoculation method and nitrogen application on each index. The different letters indicate the significance within the same year at the 5% level by the LSD test. ns, not significant, (*p >* 0.05); *, significant at *p <* 0.05; **, significant at *p <* 0.01.

	2021			2022		
	N_CL_	N_LL_	N_RL_	N_CL_	N_LL_	N_RL_
RN3	5.81 a	5.79 a	5.59 a	5.70 a	5.62 a	5.47 a
RN2	5.55 a	5.38 a	5.07 a	5.33 b	5.25 b	4.98 bc
RN1	5.16 ab	5.07 ab	5.01 a	4.97 d	4.82 c	4.59 d
RN0	4.09 bc	3.95 bc	3.82 b	3.48 f	3.41 e	3.37 e
N3	4.96 abc	4.88 abc	4.66 ab	5.44 b	5.34 b	5.19 b
N2	5.35 a	5.19 a	5.03 a	5.13 c	5.01 c	4.88 c
N1	4.98 abc	4.87 abc	4.73 ab	4.69 e	4.58 d	4.49 d
N0	3.84 c	3.72 c	3.64 b	3.13 g	3.06 f	2.98 f
Significance level						
N	*	**	**	**	**	**
R	ns	*	ns	**	*	**
N*R	ns	*	ns	*	ns	*

**Table 3 plants-13-00140-t003:** The correlation coefficients between the LNC and spectral parameters of soybean leaves. *, the correlation coefficient reached a significant level (*p* < 0.05, the same below). The bold represents the highest correlation coefficients.

Selected Spectral Parameter	Correlation Coefficient		
	N_CL_	N_LL_	N_RL_
Maximum first-order derivative value in the blue edge (490–530 nm) *D_b_*	**0.601 ***	0.582 *	0.514 *
Maximum first-order derivative value in the yellow edge (462–642 nm) *D_y_*	**0.601 ***	0.582 *	0.514 *
Maximum first-order derivative value in the red edge (670–760 nm) *D_r_*	**0.660 ***	0.586 *	0.526 *
Maximum reflectivity of the green peak (510–560 nm) *R_g_*	**0.495 ***	0.501 *	0.460 *
Minimum reflectivity of the red valley (650–690 nm) *R_r_*	**0.208**	0.259	0.258
Blue edge (490–530 nm) area *S_b_*	**0.483 ***	0.511 *	0.476 *
Yellow edge (462–642 nm) area *S_y_*	**0.176**	0.255	0.260
Red edge (670–760 nm) area *S_r_*	**0.669 ***	0.604 *	0.543 *
NDDr.b	**0.069**	0.013	0.007
NDSDr.b	**0.114**	0.015	0.0004
IPVI	**0.667 ***	0.630 *	0.582 *
Optimized soil-adjusted vegetation index (OSAVI)	**0.512 ***	0.419 *	0.369 *
Normalized difference nitrogen index (NDNI)	**0.550 ***	0.480 *	0.441 *
Ashburn vegetation index (AVI)	**0.670 ***	0.631 *	0.574 *
Difference 678/500 (D678/500)	**0.580 ***	0.510 *	0.439 *
Difference 800/550 (D800/550)	**0.664 ***	0.603 *	0.547 *
Difference 800/680 (D800/680)	**0.670 ***	0.605 *	0.544 *
Difference 833/658 (D833/658)	**0.671 ***	0.609 *	0.547 *
Differenced vegetation index MSS (DVIMSS)	**0.669 ***	0.631 *	0.574 *
Double difference index (DDI)	**0.449 ***	0.363 *	0.362 *

**Table 4 plants-13-00140-t004:** The maximum correlation coefficients and wavelength positions between the spectral indices screened by any two bands and the LNCs of different leaf layers. *, the correlation coefficient reached a significant level (*p* < 0.05, the same below). The bold represents the highest correlation coefficients.

Selected Spectral Parameter	N_CL_		N_LL_		N_RL_	
	Correlation Coefficient	Wavelength Position (*i,j*)	Correlation Coefficient	Wavelength Position (*i,j)*	Correlation Coefficient	Wavelength Position (*i,j*)
RI	**0.664 ***	(840,843)	0.567 *	(1043,1046)	0.574 *	(839,844)
DI	**0.701 ***	(1625,1637)	0.684 *	(1221,1267)	0.701 *	(1612,1611)
SAVI	**0.675 ***	(413,934)	0.662 *	(1358,1393)	0.638 *	(1611,1612)
NDVI	**0.664 ***	(840,843)	0.599 *	(1043,1045)	0.574 *	(844,839)
TVI	**0.689 ***	(694,616)	0.661 *	(1353,680)	0.594 *	(1129,487)
mSR	**0.671 ***	(840,843)	0.597 *	(1043,1045)	0.584 *	(844,839)
mNDI	**0.670 ***	(840,841)	0.597 *	(1043,1045)	0.584 *	(844,839)
PI	**0.676 ***	(856,854)	0.642 *	(753,1356)	0.593 *	(1046,1047)
SI	**0.686 ***	(783,1368)	0.663 *	(840,1368)	0.579 *	(759,1129)
VI6	**0.732 ***	(841,842)	0.658 *	(972,981)	0.639 *	(840,843)

## Data Availability

Data are contained within the article.

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
