# Peer review of "Monitoring of Nitrogen Concentration in Soybean Leaves at Multiple Spatial Vertical Scales Based on Spectral Parameters"

_plants, 2024, doi:10.3390/plants13010140_

Round 1
Reviewer 1 Report
Comments and Suggestions for Authors
This is a prospective study that analyzed the monitoring of nitrogen concentration in soybean leaves (LNC) as a key indicator for assessing plant growth and development. Efficient and non-destructive estimation of crop LNC using the Spectral technology it's very useful in crop and other plant research. The intraindividual variability of LNC on the three layers of soybean plants (the root (NRL), lateral leaf (NLL) and canopy (NCL)) were investigated. Four optimal strategies for selecting the spectral index were constructed based on the correlation coefficients between the spectral parameters and the LNC. The highest values of correlation coefficients were statistically significant in the upper part of soybean, which also showed the highest accuracy in the assessment.
This Study deserves to be published, but after a major revision.
Major revision
1. Test design
"The experiment set four nitrogen application levels (Table 1): 0 kg ha-1 (N0), 60 126 kg ha-1 (N1), 120 kg ha-1 (N2), 180 kg ha-1 (N3), and two seed dressing treatments: Rhizobium inoculation (R) and water seed dressing (unmarked)”.
The paper does not mention anywhere on which sample the LNC spectral index assessment of these mentioned levels of nitrogen concentration in the soil and in combination with Rhizobium inoculation was done.
I think it would be particularly interesting to conduct tests that would show the significance of the differences in nitrogen content in treatments (N0,N1,N2, N3 and RN0,RN1,RN2,RN3) and soybean LNC. This would result in a pattern of LNC content depending on the concentration in the soil as well as the concentration of nitrogen and the presence of Rhizobium. Only the data of the estimated values of the canopy layer of the plant can be compared, which were shown to have the highest correlation with the spectral indices and precision.
Show the reaction norms of spectral index values and LNC for all mentioned treatments. Perform ANOVA with factors of nitrogen concentration and Rhizobium inoculation treatment. The statistical significance of LNC differences within the nitrogen content treatments in the substrate (N0,N1,N2, N3 vs. RN0,RN1,RN2, RN3) as well as between treatments with and without Rhizobium (N0 vs. RN0, N1 vs. RN1, N2 vs. RN2, N3 vs, RN3), will be valuation.
This would contribute to the quality of this manuscript and enable the use the spectral methodology in other areas of plants research.
2. Missing Figure 5
3. Figures 4, 5 and 6 show correlations of spectral indices and soybean LNC according to the indices for each of the plant layers. Combining these three figures into one, in this way, the differences in the correlations of the spectral indices and plant parts will be perceived at the same time.
E.g:
Figure 4: The correlation matrix diagram of spectral index and soybean RI (a1, a2, a3), DI (b1, b2, b3), SAVI (c1, c2, c3), NDVI (d1, d2, d3), TVI ( e1, e2, e3), mSR (f1, f2, f3), mNDI (g1, g2, g3), PI (h1, h2, h3), SI (i1, i2, i3) and VI6 (j1, j2, j3 ). Number 1, LNCCL, 2. LNCLL and 3. LNCRL; Color from blue to red represents negative correlation to positive correlation.
4. Figures 1. Correction of the legend. Unacceptable "study area".
5. Delete the unnecessary table 1
6. Table 2 supplementary material
7. Figure 3. Correction of figure legend, which parameters of statistics, mean value, median, etc.
Use the same color for different plant layers.
For example:
LNCRL modeling - dark brown, visualization light brown.
LNCLL modeling - dark green, visualization light green.
LNCCL modeling - dark blue, visualization light blue.
And explanations put in the figure legends not in the graph.
8.Table 3.
Db and Dy the same values?
Then bold the highest correlation coefficient between plant layers.
For example for Db bold 0.601……….
9. Table 4.
bold the highest correlation coefficient between plant layers.
For example for RI bold 0.664……….
10. Figure 7,8,9 reorganize according the suggestions for figure 4, 5, 6.
For example
Figure 7. The BPNN combine1, 2, 3, 4 for LNCCL, LNCLL and LNCRL
Figure 8. The PLSR combine1, 2, 3, 4 for LNCCL, LNCLL and LNCRL
Figure 9. The RF combine1, 2, 3, 4 for LNCCL, LNCLL and LNCRL
In figure legend explain red and blue dat
Author Response
Reviewer #1 comments:
This is a prospective study that analyzed the monitoring of nitrogen concentration in soybean leaves (LNC) as a key indicator for assessing plant growth and development. Efficient and non-destructive estimation of crop LNC using the Spectral technology it's very useful in crop and other plant research. The intraindividual variability of LNC on the three layers of soybean plants (the root (NRL), lateral leaf (NLL) and canopy (NCL)) were investigated. Four optimal strategies for selecting the spectral index were constructed based on the correlation coefficients between the spectral parameters and the LNC. The highest values of correlation coefficients were statistically significant in the upper part of soybean, which also showed the highest accuracy in the assessment.
This Study deserves to be published, but after a major revision.
Response: Dear reviewer, thank you for your meticulous review and constructive feedback on our manuscript. We appreciate the time and effort you and the first-round reviewers have invested in evaluating our work. We are grateful for your positive assessment of the comprehensive responses and revisions made by the authors. We have now incorporated your comments and suggestions in preparation of the revised manuscript.
- Test design
"The experiment set four nitrogen application levels (Table 1): 0 kg ha-1 (N0), 60 126 kg ha-1 (N1), 120 kg ha-1 (N2), 180 kg ha-1 (N3), and two seed dressing treatments: Rhizobium inoculation (R) and water seed dressing (unmarked)”.
The paper does not mention anywhere on which sample the LNC spectral index assessment of these mentioned levels of nitrogen concentration in the soil and in combination with Rhizobium inoculation was done.
Response: Thank you for your pointing out, we set different nitrogen levels and rhizobium inoculation levels in order to make the nitrogen content of soybean different. The LNC of all plots was treated as a test sample of the spectrum.
L207: Firstly, the correlation between the spectral parameters of soybean leaves and LNC in all 24 plots was analyzed.
I think it would be particularly interesting to conduct tests that would show the significance of the differences in nitrogen content in treatments (N0, N1, N2, N3 and RN0, RN1, RN2, RN3) and soybean LNC. This would result in a pattern of LNC content depending on the concentration in the soil as well as the concentration of nitrogen and the presence of Rhizobium. Only the data of the estimated values of the canopy layer of the plant can be compared, which were shown to have the highest correlation with the spectral indices and precision.
Show the reaction norms of spectral index values and LNC for all mentioned treatments. Perform ANOVA with factors of nitrogen concentration and Rhizobium inoculation treatment. The statistical significance of LNC differences within the nitrogen content treatments in the substrate (N0, N1, N2, N3 vs. RN0, RN1, RN2, RN3) as well as between treatments with and without Rhizobium (N0 vs. RN0, N1 vs. RN1, N2 vs. RN2, N3 vs, RN3), will be valuation.
This would contribute to the quality of this manuscript and enable the use the spectral methodology in other areas of plants research.
Response: Thank you for your pointing out, we have listed the LNC differences of different treatments.
L241-246: From Table 2, it is evident that different nitrogen application rates significantly im-pact the LNC of soybeans (P < 0.05). In the majority of leaves, the inoculation of rhizobia also exerts a significant effect on LNC (P < 0.05). Notably, as nitrogen application increases, leaf LNC increases accordingly. Furthermore, under the condition of the same nitrogen application rate, rhizobia inoculation promotes nitrogen absorption in the leaves, thereby augmenting LNC.
Table 2. The value of LNC in each leaf layer under different treatments. R: the influence of different rhizobium inoculation method on each index; N: the effect of different nitrogen application on each index; Y*M: the influence of the interaction between rhizobium inoculation method and nitrogen application on each index. Different alphabets indicate the significance within the same year at 5% level by LSD test. NS: not significant, (P > 0.05); * , Significant at P < 0.05; * *, Significant at P < 0.01.
|
Treatments |
2021 |
2022 |
||||
|
NCL |
NLL |
NRL |
NCL |
NLL |
NRL |
|
|
RN3 |
5.81a |
5.79a |
5.59a |
5.70a |
5.62a |
5.47a |
|
RN2 |
5.55a |
5.38a |
5.07a |
5.33b |
5.25b |
4.98bc |
|
RN1 |
5.16ab |
5.07ab |
5.01a |
4.97d |
4.82c |
4.59d |
|
RN0 |
4.09bc |
3.95bc |
3.82b |
3.48f |
3.41e |
3.37e |
|
N3 |
4.96abc |
4.88abc |
4.66ab |
5.44b |
5.34b |
5.19b |
|
N2 |
5.35a |
5.19a |
5.03a |
5.13c |
5.01c |
4.88c |
|
N1 |
4.98abc |
4.87abc |
4.73ab |
4.69e |
4.58d |
4.49d |
|
N0 |
3.84c |
3.72c |
3.64b |
3.13g |
3.06f |
2.98f |
|
Significance level |
||||||
|
N |
* |
** |
** |
** |
** |
** |
|
R |
ns |
* |
ns |
** |
* |
** |
|
N*R |
ns |
* |
ns |
* |
ns |
* |
L361-366:We observed that the increase in nitrogen application leads to a corresponding rise in LNC. This is attributed to the heightened nitrogen application, which enhances soybean's capacity to absorb nitrogen, consequently elevating LNC[7]. Simultaneously, our findings indicate that inoculation with rhizobia enhances soybean LNC. This is attributed to the symbiotic relationship between rhizobia and soybeans, wherein rhizobia facilitate nitrogen fixation, thereby promoting nitrogen absorption[52].
- Missing Figure 5
Response: Thank you for your pointing out, we have listed Figure 5, then it is merged into Figure 4.
- Figures 4, 5 and 6 show correlations of spectral indices and soybean LNC according to the indices for each of the plant layers. Combining these three figures into one, in this way, the differences in the correlations of the spectral indices and plant parts will be perceived at the same time.
E.g:
Figure 4: The correlation matrix diagram of spectral index and soybean RI (a1, a2, a3), DI (b1, b2, b3), SAVI (c1, c2, c3), NDVI (d1, d2, d3), TVI ( e1, e2, e3), mSR (f1, f2, f3), mNDI (g1, g2, g3), PI (h1, h2, h3), SI (i1, i2, i3) and VI6 (j1, j2, j3 ). Number 1, LNCCL, 2. LNCLL and 3. LNCRL; Color from blue to red represents negative correlation to positive correlation.
Response: Thank you for your pointing out, we have merged them into Figure 4 as required.
Figure 4. The correlation matrix diagram of spectral index and soybean LNC. (a1) RI and NCL; (a2) RI and NLL; (a3) RI and NRL; (b1) DI and NCL; (b2) DI and NLL; (b3) DI and NRL; (c1) SAVI and NCL; (c2) SAVI and NLL; (c3) SAVI and NRL; (d1) NDVI and NCL; (d2) NDVI and NLL; (d3) NDVI and NRL; (e1) TVI and NCL; (e2) TVI and NLL; (e3) TVI and NRL; (f1) mSR and NCL; (f2) mSR and NLL; (f3) mSR and NRL; (g1) mNDI and NCL; (g2) mNDI and NLL; (g3) mNDI and NRL; (h1) PI and LNCCL; (h2) PI and LNCLL; (h3) PI and LNCRL; (i1) SI and LNCCL; (i2) SI and LNCLL; (i3) SI and LNCRL; (j1) VI6 and LNCCL; (j2) VI6 and LNCLL; (j3) VI6 and LNCRL. Color from blue to red represents negative correlation to positive correlation.
- Figures 1. Correction of the legend. Unacceptable "study area".
Response: Thank you for your pointing out, we have revised according to the requirements.
Figure 1. study area.
- Delete the unnecessary table 1
Response: Thank you for your pointing out, we have revised according to the requirements.
- Table 2 supplementary material
Response: Response: Thank you for your pointing out, we have supplemented the contents of Table 1 (Table 2 of the previous draft) as required.
Table 1. Selection of spectral parameters, calculation formula and reference source.
|
Selected spectra parameters |
Calculation formula |
Reference |
|
The maximum first-order derivative value in the blue edge (490-530nm) Db |
/ |
[30] |
|
The maximum first-order derivative value in the yellow edge (462-642nm) Dy |
/ |
[30] |
|
The maximum first-order derivative value in the red edge (670-760nm) Dr |
/ |
[31] |
|
The maximum reflectivity of green peak (510-560nm) Rg |
/ |
[32] |
|
The minimum reflectivity of red valley (650-690 nm) Rr |
/ |
[32] |
|
Blue edge (490-530 nm) area Sb |
The sum of first-order derivatives within the blue-edge wavelength range. |
[30] |
|
Yellow edge (462-642 nm) area Sy |
The sum of first-order derivatives within the yellow-edge wavelength range. |
[30] |
|
Red edge (670-760 nm) area Sr |
The sum of first-order derivatives within the red-edge wavelength range. |
[31] |
|
Normalized red-blue amplitude difference (NDDr.b) |
(Dr-Db)/(Dr + Db) |
[33] |
|
Normalized first-order red-blue amplitude difference (NDSDr.b) |
(SDr-SDb)/(SDr + SDb) |
[33] |
|
Infrared percentage vegetation index (IPVI) |
R800*(R800+R670) |
[34] |
|
Optimized soil-adjusted vegetation Index (OSAVI) |
[35] |
|
|
Normalized Difference Nitrogen Index (NDNI) |
[36] |
|
|
Ashburn vegetation Index (AVI) |
[37] |
|
|
Difference 678/500 (D678/500) |
[38] |
|
|
Difference 800/550 (D800/550) |
[39] |
|
|
Difference 800/680 (D800/680) |
[40] |
|
|
Difference 833/658 (D833/658) |
[41] |
|
|
Differenced Vegetation Index MSS (DVIMSS) |
[42] |
|
|
Double Difference Index (DD) |
[43] |
|
|
Ratio Index (RI) |
|
[27] |
|
Difference Index (DI) |
- |
[27] |
|
Soil-Adjusted Vegetation Index (SAVI) |
[28] |
|
|
Normalized Difference Vegetation Index (NDVI) |
[28] |
|
|
Triangular Vegetation Index (TVI) |
|
[28] |
|
Modified Simple Ratio (mSR) |
[28] |
|
|
Modified Normalized Difference Index (mNDI) |
-2 |
[28] |
|
Product index (PI) |
[44] |
|
|
Sum index (SI) |
[44] |
|
|
Reciprocal difference index (VI6) |
1/- |
[44] |
Note: (i=1,2,3) is the reflectivity at any band,R445、R455、R500、R530、R531、R550、R570、R670、R680、R700、R705、R742、R750、R800 represent the spectral reflectance of wavelength 445nm、455nm、500nm、530nm、531nm、550 nm、570 nm、670 nm、680nm、700 nm、705 nm、742 nm、750 nm and 800 nm.
- Figure 3. Correction of figure legend, which parameters of statistics, mean value, median, etc.
Use the same color for different plant layers.
For example:
LNCRL modeling - dark brown, visualization light brown.
LNCLL modeling - dark green, visualization light green.
LNCCL modeling - dark blue, visualization light blue.
And explanations put in the figure legends not in the graph.
Response: Thank you for your pointing out, we have revised according to the requirements.
Figure 3. Statistics of LNC in each leaf layer of soybean. The horizontal line in the box line diagram represents the median, and the white box represents the average value. NRL modeling set-dark brown, validation set-light brown. NLL modeling set-dark purple, validation set-light purple. NCL modeling set-dark blue, validation set-light blue.
8.Table 3.
Db and Dy the same values?
Then bold the highest correlation coefficient between plant layers.
For example for Db bold 0.601……….
Response: Thank you for your pointing out. There are repeated parts (490-530 nm) in the calculation of these two parameters, and the highest correlation coefficient appears at the same wavelength position, so the correlation coefficients of these two parameters are the same. Other places have been thickened as required.
- Table 4.
bold the highest correlation coefficient between plant layers.
For example for RI bold 0.664……….
Response: Thank you for your pointing out, the correlation coefficient has been thickened as required.
- Figure 7,8,9 reorganize according the suggestions for figure 4, 5, 6.
For example
Figure 7. The BPNN combine1, 2, 3, 4 for LNCCL, LNCLL and LNCRL
Figure 8. The PLSR combine1, 2, 3, 4 for LNCCL, LNCLL and LNCRL
Figure 9. The RF combine1, 2, 3, 4 for LNCCL, LNCLL and LNCRL
In figure legend explain red and blue date
Response: Thank you for your pointing out, we have changed Figure 7-9 to Figure 5-7.
Figure 5. The modeling set and validation set of BPNN estimation models with different input variables and leaf layers. The red dot red line represents the modeling set and the modeling set fitting curve, the blue dot and the blue line represent the verification set and the verification set fitting curve, and the dotted line represents the 1 : 1 line.
Figure 6. The modeling set and validation set of PLSR estimation models with different input variables and leaf layers. The red dot red line represents the modeling set and the modeling set fitting curve, the blue dot and the blue line represent the verification set and the verification set fitting curve, and the dotted line represents the 1 : 1 line.
Figure 7. The modeling set and validation set of RF estimation models with different input variables and leaf layers. The red dot red line represents the modeling set and the modeling set fitting curve, the blue dot and the blue line represent the verification set and the verification set fitting curve, and the dotted line represents the 1 : 1 line.

Reviewer 2 Report
Comments and Suggestions for Authors
Review: “Monitoring of Nitrogen Concentration in Soybean Leaves at Multiple Spatial Vertical Scales Based on Spectral Parameters”
The research presented in submitted manuscript focuses on estimating nitrogen concentration values in different layers of soybean leaves using non-destructive and rapid modern drone-based testing methods and the construction of three categories of spectral parameters for estimating LNC (leaf nitrogen concentration) in different layers of soybean leaves: 1) empirical spectral indices with good correlations with crop parameters from previous studies; 2) optimal spectral indices, i.e. the best combination indices in the wavelength range of 350-1 830 nm with the highest correlation with LNC in different layers of soybean leaves; and 3) triangular spectral parameters.
To obtain the most optimal results for estimating nitrogen concentration values, the authors used three machine learning models: partial least squares regression (PLSR), random forest (RF) and back propagation neural network (BPNN) algorithm.
The use of non-invasive methods to assess the nitrogen content of various parts of crops is valuable not only from a practical point of view, but above all, it will help minimize the risk of over-fertilization, thereby mitigating the impact on the environment. In my opinion for this reason, the submitted manuscript is of great importance and should be published after moderate revision.
Remarks:
1. Line 33: Explain the abbreviation VI6.
2. Line 36: Explain the abbreviation R2.
3. Line 105: It is necessary to clarify the meaning of the colors (blue, yellow and red) of the edge areas in the spectral parameters.
4. Table 2. Explain the abbreviation: NDDr.b, NDSDr.b, IPVI, PI, SI, VI6 SDr, SDb, Rj.
5. Line 215: Explain the abbreviation SMC.
6. Table 3: I recommend to unify terms concerning to leaf layers in relation to correlation coefficiernt. In Fig. 2 and in the text the following terms are used: root leaf nitrogen concentration (NRL), lateral leaf nitrogen concentration (NLL) and canopy leaf nitrogen concentration (NCL).
7. Figure 5: Figure 5 is missing from the manuscript.
8. Table 4: I recommend standardizing the terms for the leaf layers. The comment is similar to that in Table 3.
9. Line 274: I recommend standardizing the font of the LNC abbreviation. Throughout the text, all letters of the abbreviation are capitalized.
10. Figure 7 - 9: It is necessary to explain the colors on the graphs.
Author Response
Reviewer #2 comments:
The research presented in submitted manuscript focuses on estimating nitrogen concentration values in different layers of soybean leaves using non-destructive and rapid modern drone-based testing methods and the construction of three categories of spectral parameters for estimating LNC (leaf nitrogen concentration) in different layers of soybean leaves: 1) empirical spectral indices with good correlations with crop parameters from previous studies; 2) optimal spectral indices, i.e. the best combination indices in the wavelength range of 350-1 830 nm with the highest correlation with LNC in different layers of soybean leaves; and 3) triangular spectral parameters.
To obtain the most optimal results for estimating nitrogen concentration values, the authors used three machine learning models: partial least squares regression (PLSR), random forest (RF) and back propagation neural network (BPNN) algorithm.
The use of non-invasive methods to assess the nitrogen content of various parts of crops is valuable not only from a practical point of view, but above all, it will help minimize the risk of over-fertilization, thereby mitigating the impact on the environment. In my opinion for this reason, the submitted manuscript is of great importance and should be published after moderate revision.
Response: Dear reviewer, thank you for your meticulous review and constructive feedback on our manuscript. We appreciate the time and effort you and the first-round reviewers have invested in evaluating our work. We are grateful for your positive assessment of the comprehensive responses and revisions made by the authors. We have now incorporated your comments and suggestions in preparation of the revised manuscript.
Remarks:
- Line 33: Explain the abbreviation VI6.
Response: Thank you for your pointing out, we have revised according to the requirements.
L33: VI6 (reciprocal difference index)
- Line 36: Explain the abbreviation R2.
Response: Thank you for your pointing out, we have revised according to the requirements.
L37:coefficient of determination (R2)
- Line 105: It is necessary to clarify the meaning of the colors (blue, yellow and red) of the edge areas in the spectral parameters.
Response: Thank you for your pointing out, we have revised according to the requirements.
L106-108: (involving the blue, yellow, and red edge areas were considered. These parameters, often associated with the red edge, blue edge, and green edge, provide valuable insights into the spectral characteristics of the studied vegetation)
- Table 2. Explain the abbreviation: NDDr.b, NDSDr.b, IPVI, PI, SI, VI6 SDr, SDb, Rj.
Response: Thank you for your pointing out, we have revised according to the requirements.
Table 1. Selection of spectral parameters, calculation formula and reference source.
|
Selected spectra parameters |
Calculation formula |
Reference |
|
The maximum first-order derivative value in the blue edge (490-530nm) Db |
/ |
[30] |
|
The maximum first-order derivative value in the yellow edge (462-642nm) Dy |
/ |
[30] |
|
The maximum first-order derivative value in the red edge (670-760nm) Dr |
/ |
[31] |
|
The maximum reflectivity of green peak (510-560nm) Rg |
/ |
[32] |
|
The minimum reflectivity of red valley (650-690 nm) Rr |
/ |
[32] |
|
Blue edge (490-530 nm) area Sb |
The sum of first-order derivatives within the blue-edge wavelength range. |
[30] |
|
Yellow edge (462-642 nm) area Sy |
The sum of first-order derivatives within the yellow-edge wavelength range. |
[30] |
|
Red edge (670-760 nm) area Sr |
The sum of first-order derivatives within the red-edge wavelength range. |
[31] |
|
Normalized red-blue amplitude difference (NDDr.b) |
(Dr-Db)/(Dr + Db) |
[33] |
|
Normalized first-order red-blue amplitude difference (NDSDr.b) |
(SDr-SDb)/(SDr + SDb) |
[33] |
|
Infrared percentage vegetation index (IPVI) |
R800*(R800+R670) |
[34] |
|
Optimized soil-adjusted vegetation Index (OSAVI) |
[35] |
|
|
Normalized Difference Nitrogen Index (NDNI) |
[36] |
|
|
Ashburn vegetation Index (AVI) |
[37] |
|
|
Difference 678/500 (D678/500) |
[38] |
|
|
Difference 800/550 (D800/550) |
[39] |
|
|
Difference 800/680 (D800/680) |
[40] |
|
|
Difference 833/658 (D833/658) |
[41] |
|
|
Differenced Vegetation Index MSS (DVIMSS) |
[42] |
|
|
Double Difference Index (DD) |
[43] |
|
|
Ratio Index (RI) |
|
[27] |
|
Difference Index (DI) |
- |
[27] |
|
Soil-Adjusted Vegetation Index (SAVI) |
[28] |
|
|
Normalized Difference Vegetation Index (NDVI) |
[28] |
|
|
Triangular Vegetation Index (TVI) |
|
[28] |
|
Modified Simple Ratio (mSR) |
[28] |
|
|
Modified Normalized Difference Index (mNDI) |
-2 |
[28] |
|
Product index (PI) |
[44] |
|
|
Sum index (SI) |
[44] |
|
|
Reciprocal difference index (VI6) |
1/- |
[44] |
Note: (i=1,2,3) is the reflectivity at any band,R445、R455、R500、R530、R531、R550、R570、R670、R680、R700、R705、R742、R750、R800 represent the spectral reflectance of wavelength 445nm、455nm、500nm、530nm、531nm、550 nm、570 nm、670 nm、680nm、700 nm、705 nm、742 nm、750 nm and 800 nm.
- Line 215: Explain the abbreviation SMC.
This is our mistake. The mistake LNC described as SMC has now been corrected.
Response: Thank you for your pointing out, we have revised according to the requirements.
- Table 3: I recommend to unify terms concerning to leaf layers in relation to correlation coefficiernt. In Fig. 2 and in the text the following terms are used: root leaf nitrogen concentration (NRL), lateral leaf nitrogen concentration (NLL) and canopy leaf nitrogen concentration (NCL).
Response: Thank you for your pointing out, we have unified the terms NRL, NLL and NCL.
Table 3. The correlation coefficient between LNC and spectral parameters of soybean leaves. * indicates that the correlation coefficient reaches a significant level (P < 0.05, the same below). The thickening represents the highest correlation coefficient.
|
Selected spectra parameters |
Correlation Coefficient |
||
|
NCL |
NLL |
NRL |
|
|
The maximum first-order derivative value in the blue edge (490-530nm) Db |
0.601* |
0.582* |
0.514* |
|
The maximum first-order derivative value in the yellow edge (462-642nm) Dy |
0.601* |
0.582* |
0.514* |
|
The maximum first-order derivative value in the red edge (670-760nm) Dr |
0.660* |
0.586* |
0.526* |
|
The maximum reflectivity of green peak(510-560nm) Rg |
0.495* |
0.501* |
0.460* |
|
The minimum reflectivity of red valley (650-690 nm) Rr |
0.208 |
0.259 |
0.258 |
|
Blue edge (490-530 nm) area Sb |
0.483* |
0.511* |
0.476* |
|
Yellow edge (462-642 nm) area Sy |
0.176 |
0.255 |
0.260 |
|
Red edge (670-760 nm) area Sr |
0.669* |
0.604* |
0.543* |
|
NDDr.b |
0.069 |
0.013 |
0.007 |
|
NDSDr.b |
0.114 |
0.015 |
0.0004 |
|
IPVI |
0.667* |
0.630* |
0.582* |
|
Optimized soil-adjusted vegetation Index (OSAVI) |
0.512* |
0.419* |
0.369* |
|
Normalized Difference Nitrogen Index (NDNI) |
0.550* |
0.480* |
0.441* |
|
Ashburn vegetation Index (AVI) |
0.670* |
0.631* |
0.574* |
|
Difference 678/500 (D678/500) |
0.580* |
0.510* |
0.439* |
|
Difference 800/550 (D800/550) |
0.664* |
0.603* |
0.547* |
|
Difference 800/680 (D800/680) |
0.670* |
0.605* |
0.544* |
|
Difference 833/658 (D833/658) |
0.671* |
0.609* |
0.547* |
|
Differenced Vegetation Index MSS (DVIMSS) |
0.669* |
0.631* |
0.574* |
|
Double Difference Index (DDI) |
0.449* |
0.363* |
0.362* |
- Figure 5: Figure 5 is missing from the manuscript.
Response: Thank you for your pointing out, we have listed Figure 5, then it is merged into Figure 4.
- Table 4: I recommend standardizing the terms for the leaf layers. The comment is similar to that in Table 3.
Response: Thank you for your pointing out, we have revised according to the requirements.
Table 4. The maximum correlation coefficient and wavelength position between the spectral index screened by any two bands and the LNC of different leaf layers. * indicates that the correlation coefficient reaches a significant level (P < 0.05, the same below). The thickening represents the highest correlation coefficient.
|
Selected spectra parameters |
NCL |
NLL |
NRL |
|||
|
Correlation Coefficient |
Wavelength position(i,j) |
Correlation Coefficient |
Wavelength position(i,j) |
Correlation Coefficient |
Wavelength position(i,j) |
|
|
RI |
0.664* |
(840,843) |
0.567* |
(1043,1046) |
0.574* |
(839,844) |
|
DI |
0.701* |
(1625,1637) |
0.684* |
(1221,1267) |
0.701* |
(1612,1611) |
|
SAVI |
0.675* |
(413,934) |
0.662* |
(1358,1393) |
0.638* |
(1611,1612) |
|
NDVI |
0.664* |
(840,843) |
0.599* |
(1043,1045) |
0.574* |
(844,839) |
|
TVI |
0.689* |
(694,616) |
0.661* |
(1353,680) |
0.594* |
(1129,487) |
|
mSR |
0.671* |
(840,843) |
0.597* |
(1043,1045) |
0.584* |
(844,839) |
|
mNDI |
0.670* |
(840,841) |
0.597* |
(1043,1045) |
0.584* |
(844,839) |
|
PI |
0.676* |
(856,854) |
0.642* |
(753,1356) |
0.593* |
(1046,1047) |
|
SI |
0.686* |
(783,1368) |
0.663* |
(840,1368) |
0.579* |
(759,1129) |
|
VI6 |
0.732* |
(841,842) |
0.658* |
(972,981) |
0.639* |
(840,843) |
- Line 274: I recommend standardizing the font of the LNC abbreviation. Throughout the text, all letters of the abbreviation are capitalized.
Response: Thank you for your pointing out, we have revised according to the requirements.
L300: Construction of Soybean LNC Estimation Model at Multi-Spatial Vertical Scale
- Figure 7 - 9: It is necessary to explain the colors on the graphs.
Response: Thank you for your pointing out, necessary explanations have been added, from blue to red represents a negative correlation to a positive correlation.

Reviewer 3 Report
Comments and Suggestions for Authors
The manuscript is very well prepared and presented. Some minor omissions are noted.
1. Materials and Methods
Line 167: It seems that Field spec3 hyperspectrometer is a portable device, and it would be better to be given the accuracy and precision of the instrument.
Line 194: All used abbreviations should be listed below the table.
2. Results and Discussion:
Lines 272-274: In Table 4 are given the maximum correlation coefficients between the spectral indexes and the LNC of different layers. Are all these correlations statistically significant?
The novelty of this study should be clearly explained. The strengths and the limitations of the study should be specified.
Overall Recommendation
The paper can in principle be accepted after minor revision.
Comments on the Quality of English LanguageMinor editing required.
Author Response
Reviewer #3 comments:
The manuscript is very well prepared and presented. Some minor omissions are noted.
- Materials and Methods
Line 167: It seems that Field spec3 hyperspectrometer is a portable device, and it would be better to be given the accuracy and precision of the instrument.
Response: Thank you for your pointing out, we have revised according to the requirements.
L171-176: The wavelength range of the instrument is 350 ~ 1 830 nm. The spectral resolution of 350 ~ 1 000 nm is 3 nm, and the sampling interval is 1.4 nm. The resolution of 1 000 ~ 1 830 nm is 10 nm, and the sampling interval is 2 nm. The instrument automatically interpolates the sampling data into 1 nm interval output. The fiber length is 1.5 m and the field of view is 25 °. The determination was carried out at 11 : 00-13 : 00 in sunny and windless weather.
Line 194: All used abbreviations should be listed below the table.
Response: Thank you for your pointing out, the full name and abbreviation have been added to the table.
Table 1. Selection of spectral parameters, calculation formula and reference source.
|
Selected spectra parameters |
Calculation formula |
Reference |
|
The maximum first-order derivative value in the blue edge (490-530nm) Db |
/ |
[30] |
|
The maximum first-order derivative value in the yellow edge (462-642nm) Dy |
/ |
[30] |
|
The maximum first-order derivative value in the red edge (670-760nm) Dr |
/ |
[31] |
|
The maximum reflectivity of green peak (510-560nm) Rg |
/ |
[32] |
|
The minimum reflectivity of red valley (650-690 nm) Rr |
/ |
[32] |
|
Blue edge (490-530 nm) area Sb |
The sum of first-order derivatives within the blue-edge wavelength range. |
[30] |
|
Yellow edge (462-642 nm) area Sy |
The sum of first-order derivatives within the yellow-edge wavelength range. |
[30] |
|
Red edge (670-760 nm) area Sr |
The sum of first-order derivatives within the red-edge wavelength range. |
[31] |
|
Normalized red-blue amplitude difference (NDDr.b) |
(Dr-Db)/(Dr + Db) |
[33] |
|
Normalized first-order red-blue amplitude difference (NDSDr.b) |
(SDr-SDb)/(SDr + SDb) |
[33] |
|
Infrared percentage vegetation index (IPVI) |
R800*(R800+R670) |
[34] |
|
Optimized soil-adjusted vegetation Index (OSAVI) |
[35] |
|
|
Normalized Difference Nitrogen Index (NDNI) |
[36] |
|
|
Ashburn vegetation Index (AVI) |
[37] |
|
|
Difference 678/500 (D678/500) |
[38] |
|
|
Difference 800/550 (D800/550) |
[39] |
|
|
Difference 800/680 (D800/680) |
[40] |
|
|
Difference 833/658 (D833/658) |
[41] |
|
|
Differenced Vegetation Index MSS (DVIMSS) |
[42] |
|
|
Double Difference Index (DD) |
[43] |
|
|
Ratio Index (RI) |
|
[27] |
|
Difference Index (DI) |
- |
[27] |
|
Soil-Adjusted Vegetation Index (SAVI) |
[28] |
|
|
Normalized Difference Vegetation Index (NDVI) |
[28] |
|
|
Triangular Vegetation Index (TVI) |
|
[28] |
|
Modified Simple Ratio (mSR) |
[28] |
|
|
Modified Normalized Difference Index (mNDI) |
-2 |
[28] |
|
Product index (PI) |
[44] |
|
|
Sum index (SI) |
[44] |
|
|
Reciprocal difference index (VI6) |
1/- |
[44] |
Note: (i=1,2,3) is the reflectivity at any band,R445、R455、R500、R530、R531、R550、R570、R670、R680、R700、R705、R742、R750、R800 represent the spectral reflectance of wavelength 445nm、455nm、500nm、530nm、531nm、550 nm、570 nm、670 nm、680nm、700 nm、705 nm、742 nm、750 nm and 800 nm.
- Results and Discussion:
Lines 272-274: In Table 4 are given the maximum correlation coefficients between the spectral indexes and the LNC of different layers. Are all these correlations statistically significant?
Response: Thank you for your pointing out, significantly related results have been added to table 4.
Table 4. The maximum correlation coefficient and wavelength position between the spectral index screened by any two bands and the LNC of different leaf layers. * indicates that the correlation coefficient reaches a significant level (P < 0.05, the same below). The thickening represents the highest correlation coefficient.
|
Selected spectra parameters |
NCL |
NLL |
NRL |
|||
|
Correlation Coefficient |
Wavelength position(i,j) |
Correlation Coefficient |
Wavelength position(i,j) |
Correlation Coefficient |
Wavelength position(i,j) |
|
|
RI |
0.664* |
(840,843) |
0.567* |
(1043,1046) |
0.574* |
(839,844) |
|
DI |
0.701* |
(1625,1637) |
0.684* |
(1221,1267) |
0.701* |
(1612,1611) |
|
SAVI |
0.675* |
(413,934) |
0.662* |
(1358,1393) |
0.638* |
(1611,1612) |
|
NDVI |
0.664* |
(840,843) |
0.599* |
(1043,1045) |
0.574* |
(844,839) |
|
TVI |
0.689* |
(694,616) |
0.661* |
(1353,680) |
0.594* |
(1129,487) |
|
mSR |
0.671* |
(840,843) |
0.597* |
(1043,1045) |
0.584* |
(844,839) |
|
mNDI |
0.670* |
(840,841) |
0.597* |
(1043,1045) |
0.584* |
(844,839) |
|
PI |
0.676* |
(856,854) |
0.642* |
(753,1356) |
0.593* |
(1046,1047) |
|
SI |
0.686* |
(783,1368) |
0.663* |
(840,1368) |
0.579* |
(759,1129) |
|
VI6 |
0.732* |
(841,842) |
0.658* |
(972,981) |
0.639* |
(840,843) |
The novelty of this study should be clearly explained. The strengths and the limitations of the study should be specified.
Response: Thank you for your pointing out, the innovative introduction is written, and the shortcomings and prospects are reflected in the discussion.
Novelty: L94-122:In such cases, using the same wavelengths may result in inadequate utilization of spectral data, limiting the effectiveness of the calculated spectral index inversion model and reducing model accuracy [28]. Furthermore, for crops, there is a lack of comparison and discussion on the prediction effects of different vertical scales of LNC. Models are often built and analyzed using LNC from a specific site without comparison and optimization, which limits the spatial applicability of the established prediction models [29].
To address these issues, this study aims to construct three categories of spectral parameters for estimating LNC at various soybean leaf layers: 1) empirical spectral indices with good correlations to crop parameters from previous studies; 2) optimal spectral indices, i.e., the best combination indices within the wavelength range of 350-1 830 nm with the highest correlation to LNC at various soybean leaf layers; and 3) three-edge spectral (involving the blue, yellow, and red edge areas were considered. These parameters, often associated with the red edge, blue edge, and green edge, provide valuable insights into the spectral characteristics of the studied vegetation) parameters, including blue, yellow, and red edge areas. The four-node stage (V4) of soybeans, occurring when the fourth true leaf unfolds after planting, is a critical period for soybean growth and development [7]. During this stage, plants begin rapid growth and establish initial structures, such as leaves and stems. The health of plants at this stage directly affects subsequent growth and development. Additionally, soybeans require sufficient nutrients to support their growth and development at this stage. Therefore, monitoring LNC during this period is of paramount importance for field management to ensure soybeans receive the necessary nutrition for stable and high yields. In this study, different soybean leaf layers' LNC under various treatments at the V4 stage are selected as the research objects. We construct different types of spectral parameters and establish soybean LNC estimation models based on the partial least squares regression (PLSR), random forest (RF), and backpropagation neural network (BPNN) algorithms. We compare and analyze the estimation effects and stability of the models, aiming to establish LNC prediction models at different vertical scales for soy-beans to achieve non-destructive and rapid LNC estimation.
The strengths and the limitations of the study: L405-414:This study aimed to estimate soybean LNC in vertical layers using spectral parameters calculated from drone multispectral image information. As soybean field management has significant practical implications, further research is needed. Future studies could consider developing more precise and accurate new indices rather than relying solely on conventional indices. Additionally, the use of drone hyperspectral imaging and thermal infrared imaging will be considered. Furthermore, spectral sensors will be adjusted multiple times in terms of angles and heights to capture more spectral information of soybean plants in three-dimensional space, achieving precise monitoring of physiological data for lower leaves and ultimately improving the estimation accuracy of soybean LNC and enhancing soybean yield.

Round 2
Reviewer 1 Report
Comments and Suggestions for Authors
Thank you for accepted my suggestions and corrected fast.
minor
Figure 1 change the legend "study area" with some mean description of locality
Figure 5, 6, 7
must be a same size
Figure legend 5
must be same paragraph format according the journal instructions